# Spectroscopic MRI-Based Biomarkers Predict Survival for Newly Diagnosed Glioblastoma in a Clinical Trial

**DOI:** 10.3390/cancers15133524

**Published:** 2023-07-07

**Authors:** Anuradha G. Trivedi, Karthik K. Ramesh, Vicki Huang, Eric A. Mellon, Peter B. Barker, Lawrence R. Kleinberg, Brent D. Weinberg, Hui-Kuo G. Shu, Hyunsuk Shim

**Affiliations:** 1Department of Radiation Oncology, Emory University School of Medicine, Atlanta, GA 30322, USA; 2Department of Biomedical Engineering, Emory University and Georgia Institute of Technology, Atlanta, GA 30332, USA; 3Department of Radiation Oncology, Sylvester Comprehensive Cancer Center, Miller School of Medicine, University of Miami, Miami, FL 45056, USA; 4Department of Radiology and Radiological Science, Johns Hopkins University, Baltimore, MD 21218, USA; 5Department of Radiation Oncology, Johns Hopkins University, Baltimore, MD 21218, USA; 6Department of Radiology and Imaging Sciences, Emory University School of Medicine, Atlanta, GA 30322, USA; 7Winship Cancer Institute, Emory University School of Medicine, Atlanta, GA 30322, USA

**Keywords:** spectroscopic MRI, survival biomarkers, glioblastoma, radiation therapy, dose-escalation

## Abstract

**Simple Summary:**

Due to the infiltrative nature of glioblastoma, standard MRI techniques, such as T1-weighted contrast-enhanced (T1w-CE) and T2-weighted fluid-attenuated inversion recovery (FLAIR), imperfectly delineate radiation-targeted tumor volume. With spectroscopic MRI, the ratio of choline elevation, a tumor biomarker and N-acetylaspartate reduction, a healthy neuronal biomarker, can better determine the extent of the tumor. The aim of our secondary analysis was to determine if there was a relationship between survival outcomes and biomarkers identified by spectroscopic MRI for a cohort of 28 glioblastoma patients who received high-dose radiation guided by spectroscopic MRI. We determined that the volume of post-surgical spectroscopically abnormal tissue was a biomarker of overall and progression-free survival, whereas the volume of residual contrast enhancement, determined by T1w-CE MRI, was not. Our results suggest that accurate delineation and treatment of an infiltrative tumor not identified by contrast is a critical component of glioblastoma management and patient survival.

**Abstract:**

Despite aggressive treatment, glioblastoma has a poor prognosis due to its infiltrative nature. Spectroscopic MRI-measured brain metabolites, particularly the choline to N-acetylaspartate ratio (Cho/NAA), better characterizes the extent of tumor infiltration. In a previous pilot trial (NCT03137888), brain regions with Cho/NAA ≥ 2x normal were treated with high-dose radiation for newly diagnosed glioblastoma patients. This report is a secondary analysis of that trial where spectroscopic MRI-based biomarkers are evaluated for how they correlate with progression-free and overall survival (PFS/OS). Subgroups were created within the cohort based on pre-radiation treatment (pre-RT) median cutoff volumes of residual enhancement (2.1 cc) and metabolically abnormal volumes used for treatment (19.2 cc). We generated Kaplan–Meier PFS/OS curves and compared these curves via the log-rank test between subgroups. For the subgroups stratified by metabolic abnormality, statistically significant differences were observed for PFS (*p* = 0.019) and OS (*p* = 0.020). Stratification by residual enhancement did not lead to observable differences in the OS (*p* = 0.373) or PFS (*p* = 0.286) curves. This retrospective analysis shows that patients with lower post-surgical Cho/NAA volumes had significantly superior survival outcomes, while residual enhancement, which guides high-dose radiation in standard treatment, had little significance in PFS/OS. This suggests that the infiltrating, non-enhancing component of glioblastoma is an important factor in patient outcomes and should be treated accordingly.

## 1. Introduction

Glioblastoma is the most common malignant primary brain tumor in need of novel treatment strategies due to its poor prognosis [1]. Despite improvements in the standard of care for glioblastoma patients, it is challenging to identify the extent of tumor infiltration with T1-weighted contrast-enhanced (T1w-CE) MRI and T2-weighted fluid-attenuated inversion recovery (FLAIR) MRI, which are currently used to determine the targets of radiation therapy [2]. Glioblastomas are characterized by a compromised blood–brain barrier and leaky vasculature, which can be identified by a gadolinium-based contrast agent in T1w-CE scans. FLAIR is used to identify tumor, edema, inflammation, and radiation effects but is not specific to tumor [3,4]. The combination of imaging methods may not identify the full extent of tumor infiltration, resulting in undertreatment during radiation therapy. The standard of care radiation therapy target is the residual contrast enhancing contour and resection cavity treated with 60 Gy with a lower dose to the surrounding FLAIR abnormality [5]. Temozolomide is administered concurrently, during and after radiation. This treatment results in a median overall survival (OS) of 16 months and median progression-free survival (PFS) of 4–7 months in historical cohorts [1,6,7].

To better identify infiltrative tumors for treatment targeting, members of our group have been developing a whole-brain, high-resolution, 3D echo-planar magnetic resonance spectroscopic imaging (MRSI) sequence that we have termed “spectroscopic MRI” [8,9,10,11]. Tumor cell metabolism differs from healthy brain tissue metabolism in that there are increased choline and reduced N-acetylaspartate levels in proliferating tumor cells. The ratio of choline to N-acetylaspartate (Cho/NAA) has been shown to be a highly specific tumor biomarker by image–histology correlation studies and predated tumor recurrence patterns [12].

We recently reported results from a multisite clinical study (NCT03137888) where 30 newly diagnosed glioblastoma patients were treated with escalated dose radiation (75 Gy), guided by the Cho/NAA abnormality on spectroscopic MRI. This treatment paradigm resulted in a median OS of 23.0 months and PFS of 16.6 months, a significant improvement from patients treated with standard of care radiation therapy dosing [13].

In this report, we perform a retrospective analysis of the data from our previously mentioned clinical study. We aim to assess spectroscopic MRI-based biomarkers of patient OS and PFS. Because historical results show that extent of resection (EOR) and residual contrast-enhancing volume are factors that correlate with survival, we hypothesize that incorporating metabolic information from spectroscopic MRI will better predict survival outcomes. We tested this using the Kaplan–Meier estimator for predicting PFS and OS and analyzed the difference in survival distributions based on the residual contrast-enhancing and spectroscopic MRI-based volumes.

## 2. Materials and Methods

### 2.1. Description of Tumor Volume Determination/Target Generation

Patients enrolled in this study had pathologically confirmed, newly diagnosed World Health Organization (WHO) grade IV glioblastoma, which at the time of enrollment, included isocitrate dehydrogenase (IDH) mutant patients. In 2021, the WHO definition of grade IV glioblastoma was changed to exclude IDH mutant patients [14]. In accordance with the updated definition, we performed our analysis excluding the two IDH mutant patients (*n* = 28). Further trial details, including patient selection and sample selection, as well as initial outcome results for all 30 enrolled patients, have been previously reported [13]. The patients in this study provided informed consent for participation according to the institutional review board at each participating institution, and each institutional review board approved this study. After surgical resection and prior to beginning radiation therapy, a spectroscopic MRI was acquired in addition to a standard MRI (T1w-CE and FLAIR MRIs). Spectroscopic MRIs were acquired on Siemens 3T scanners at each institution with an echo planar spectroscopic imaging pulse sequence with GRAPPA parallelization. Either a 20- or 32-channel head and neck coil was used with TE = 50 ms, TR = 1551 ms, and FA = 71° [13]. Scans were processed and registered to anatomic T1 MRIs for each patient before radiation treatment planning [15,16,17,18,19].

Pre-resection scans were obtained 0–18 days before surgery, and post-resection scans were obtained approximately 3–4 weeks after surgery and 1 week prior to the radiation therapy start date. For treatment planning, the contrast-enhancing regions of the pre-resection T1w-CE, including necrotic centers, were contoured. The residual post-contrast enhancement (rENH) tumor volume was semi-automatically generated on the Brain Imaging Collaboration Suite using the previously reported method by contouring only the contrast-enhancing regions of the T1w-CE scan for each patient inside of the FLAIR envelope, excluding the resection cavity [20,21]. All rENH volumes were checked and, if necessary, edited by a board certified neuroradiologist. The Brain Imaging Collaboration Suite also automatically generated a spectroscopic MRI contour for Cho/NAA ≥ 2x, which was then manually edited by site MRS experts based on spectral quality. The gross tumor volume 3 (GTV3) was calculated for each patient by combining the rENH volume with the Cho/NAA ≥ 2x volume to determine the escalated dose target. The GTV3 contours were inspected and approved by two board-certified radiation oncologists (one of them was the treating physician). This workflow was published in Gurbani et al. [21].

Briefly, for each patient enrolled, radiation treatment contours were guided by standard MRIs and spectroscopic MRI data as follows: FLAIR abnormal regions of the brain received 50.1 Gy of radiation, the resection cavity received 60 Gy, and the GTV3 contour received a boosted 75 Gy of radiation [21]. After completing radiation therapy, imaging was acquired every 2–3 months as per standard practice. This follow-up imaging was used to determine PFS, and OS was determined through chart review and communication between patients, their families, their oncology team, and published obituaries [13]. In this report, OS and PFS were calculated as of 17 months after completing enrollment of the last patient.

In Figure 1, an example patient from the completed clinical trial is shown. This patient had a gross total resection (GTR) with no residual enhancing tumor. For this patient, standard of care would have used the resection cavity with a margin for high-dose radiation (60 Gy). This figure demonstrates how, even when using a 20 mm margin, some abnormal tissue in the GTV3 volume detected by spectroscopic MRI is left out. Furthermore, most clinical practices use a margin between 5 and 10 mm, which, in this case, would have left out a significant portion of tumor that spectroscopic MRI was able to detect. It is important to note that the Dice similarity coefficients in the figure show how much of the cavity and margins overlap with the GTV3 contour. In addition, using large, homogeneous margins can lead to unnecessary treatment of healthy tissue with high-dose radiation. We calculated the percentages of volumes that would be outside of the GTV3 contour. For a 5, 10, and 20 mm expansion beyond the cavity, 41.3%, 49.7%, and 74.9% of each volume would fall outside of the GTV3 volume, respectively. While there is some overlap between standard of care high-dose targets and our GTV3 target, the target guided by spectroscopic MRI illustrates metabolically active tumors undetected by T1w-CE.

In Figure 2, two subjects are shown; Case 2 is the same patient shown in Figure 1. The vast difference in rENH and GTV3 contour volumes for the two subjects is shown here, particularly in Case 2 where the volume of residual enhancement was 0 cc. Both contours are also rendered in 3D to emphasize the spatial differences between the rENH and GTV3 contours.

### 2.2. Surgical Resection Classification

EOR was calculated based on the percent change between the pre-operative contrast-enhancing tumor volume and the rENH volume. When EOR ≥ 95%, the surgery was classified as gross total resection (GTR), while an EOR < 95% was considered subtotal resection (STR), excluding biopsy. A secondary method of surgical resection classification was also used, in which resection was classified as GTR if the rENH volume was ≤1.5 cc or STR if >1.5 cc. Surgery classification by EOR and rENH volume was used to compare the 28 patients considered in this analysis to patients in large historical cohorts [22,23,24].

### 2.3. Statistical Analysis/Survival Analysis

The Kaplan–Meier estimator was used to calculate survival curves based on OS and PFS, with survival measured as the time between surgical resection for each patient and their recorded OS and PFS dates [25]. Patients were first stratified into two groups using median rENH and median GTV3 volumes. Each Kaplan–Meier curve was compared using a log-rank test to determine differences between patients with rENH volumes greater than and less than the median rENH volume, and patients with GTV3 volumes greater than and less than the median GTV3 volume with regard to OS and PFS. Linear regression was performed and the associated R^2^ value was used to determine any correlation between the rENH and GTV3 volumes, which would confound any comparisons. Statistical analysis was conducted using the Python lifelines analysis library [26]. The secondary analysis performed in this report was approved by the institutional review boards at each institution.

## 3. Results

Of the 30 patients included in the original study, 2 had IDH mutations (IDH1 R132H) and 9 were O^6^-methylguanine-DNA methyltransferase (MGMT) hypermethylated. Details including age, IDH and MGMT statuses, resection classification, and treatment volumes for each patient can be found in Appendix A; subjects 1 and 15 were excluded from the analysis performed in this report due to their IDH mutations. As reported previously, 16 patients had died with a median OS of 23.0 months [13]. Excluding the two IDH mutant patients, the median time to follow-up for censored patients was 22.4 months (5.7–30.1 months), and the median PFS was 16.2 months, as shown on a Kaplan–Meier curve in Figure 3. The median age for the 28 patients was 59.3 years with a range of 38.1–71.6 years (minimum–maximum) and there were 10 females and 18 males. Without the two IDH mutant patients, the median OS becomes 21.6 months and the median PFS becomes 16.2 months.

The volume of high-risk disease remaining should be a predictor of survival outcomes for glioblastoma patients. EOR or rENH are classic ways to identify the amount of high-risk disease and have been shown in multiple previous studies to be correlated with both OS and PFS. Twelve out of twenty-eight patients in our cohort had GTR (defined by either EOR ≥ 95% or rENH ≤ 1.5 cc), as shown in Table 1. Using this cutoff (GTR versus less than GTR), we did not find a significant difference in survival outcomes. However, pre-treatment spectroscopic MRI scans present the opportunity to apply additional criteria for identifying high-risk disease that may be superior to rENH alone.

On our pilot study, GTV3 volume was hypothesized to provide a better representation of residual high-risk disease than rENH alone. For our cohort, median rENH was 2.1 cc (range: 0.0–15.4 cc) and median GTV3 was 19.2 cc (range: 0.9–65.0 cc). The GTV3 volumes were always greater than rENH volumes (median of 8.2 times larger). In Figure 4, a scatterplot of rENH vs. GTV3 volumes for each patient indicates a low correlation between the volumes (R^2^ = 0.127, *p* = 0.063).

Figure 5 shows Kaplan–Meier curves for OS and PFS using the rENH and GTV3 volumes as predictive markers. Figure 5A,B shows Kaplan–Meier curves for OS. In Figure 5A, the 28 patients are stratified by those with an rENH volume below and above the median (2.1 cc). Similarly, in Figure 5B, the 28 patients are stratified by those with a GTV3 volume below and above the median (19.2 cc). The difference in medians for the groups stratified by rENH is 1.7 months (*p* = 0.373), while the difference in medians for the groups stratified by GTV3 is 10.3 months (*p* = 0.020), indicating that GTV3 volume is a significant biomarker of OS. For patients with rENH < 2.1 cc, the median OS was 23.0 months; for rENH > 2.1 cc, the median OS was 21.3 months. For patients with GTV3 < 19.2 cc, the median OS was 23.0 months; for GTV3 > 19.2 cc, the median OS was 17.1 months.

Figure 5C,D shows Kaplan–Meier curves for PFS using the same predictive markers as the OS graphs. The difference in medians for the groups stratified by rENH is 6.2 months shown in Figure 5C (*p* = 0.286), while the difference in medians for the groups stratified by GTV3 is 11.4 months shown in Figure 5D (*p* = 0.019), indicating that GTV3 volume is a better predictive marker for PFS. For patients with rENH < 2.1 cc, the median PFS was 19.0 months; for rENH > 2.1 cc, the median PFS was 12.8 months. For patients with GTV3 < 19.2 cc, median PFS was 24.0 months; for GTV3 > 19.2 cc, the median PFS was 12.6 months.

## 4. Discussion

In this paper, we have used data from a clinical trial using spectroscopic MRI to guide dose-escalation therapy to glioblastoma to assess the relationship between the rENH tumor and metabolically active tumor, as detected by spectroscopic MRI, and assess survival outcomes (PFS and OS) to determine if residual spectroscopically abnormal tumor (GTV3) was a biomarker for survival. For every subject, the rENH volume was smaller than residual non-enhancing tumor that was detected using spectroscopy, since GTV3 contains both residual enhancing tumor and non-enhancing metabolically active tumor. Additionally, there was low correlation between GTV3 and rENH volumes (R^2^ = 0.127). The lack of a linear relationship between the volumes is further supported by the insignificant *p*-value (*p* = 0.063).

The amount of residual spectroscopically abnormal tumor, as measured by the GTV3 volume, was a statistically significant biomarker for both OS (*p* = 0.020) and PFS (*p* = 0.019). We hypothesize that these results are attributed to the metabolically active tumor within the GTV3 volume that is not accounted for in the rENH volume. The large gap and lack of crossover between the GTV3-stratified Kaplan–Meier curves suggest that GTV3 is a better indicator of OS and PFS than rENH in this patient cohort. For this reason, GTV3 may be a better predictor of PFS and OS because it more accurately reflects how much non-infiltrating tumor is left post-resection. It also provides more precise targeting compared to adding a 1–2 cm margin around the resection cavity for the high-dose target, allowing higher radiation doses to be used with a decreased risk of damaging healthy tissue. Unlike prior studies, there was no relationship between rENH and survival or EOR and survival [27]. This result may be attributed to the relatively low amounts of residual contrast-enhancing tumor on most patients in this study, due to aggressive resection or higher radiation doses given to these regions.

For this study, the EOR classified using rENH volume (using a threshold of 1.5 cc) or EOR percentage (using a threshold of 95%) was similar to large historical cohorts, such as RTOG 0825 (637 patients), AVAGlio (921 patients), and EF-14 (695 patients) [22,23,24]. A 2005–2010 study of 128 patients from the Cleveland Clinic reported a similar distribution based on rENH and EOR [28]. Their study resulted in a median survival of 16 months for patients with rENH ≤ 1.5 cc and a median survival between 14 and 15 months for patients with EOR ≥ 95%. Spectroscopic MRI-guided dose-escalation resulted in a median survival of 23 months for patients with rENH ≤ 1.5 cc and a median survival of 27 months for patients with EOR ≥ 95%, which are more favorable outcomes. Similar studies have been previously reported, often with factors such as GTR versus STR, diffusion, or other volumes. Haj et al. performed a survival analysis between patients that received GTR versus STR [29]. They identified a 4-month difference in OS (GTR = 18.5 months, STR = 14.5 months) and a 1.6-month difference in PFS (GTR = 9.4 months, STR = 7.8 months). Awad et al. performed a survival analysis using only EOR and a combination of EOR and pre-resection contrast-enhancing volume [30]. Their EOR-only comparison failed to identify a relationship with survival outcomes; however, their EOR/pre-resection contrast-enhancing volume comparison was identified as a significant predictor of patient survival. Compared to the studies mentioned above, the number of patients in our trial is smaller and a larger study is warranted for fairer comparisons.

Limitations of this analysis include the relatively small cohort size, a treatment paradigm which was not a clinical standard of care, and data from only three clinical sites. More data is needed to determine if the amount of spectroscopically abnormal residual tumor after resection continues to have this predictive effect in larger patient groups, but this preliminary data is supportive of further study of using the extent of metabolically abnormal tissue to predict patient outcomes.

## 5. Conclusions

Due to the infiltrative nature of glioblastoma, T1w-CE and FLAIR MRIs may not optimally characterize the extent of tumor, resulting in undertreatment during radiation therapy. Our goal was to use data from a previously performed clinical trial to determine how spectroscopic MRI-based biomarkers predicted PFS and OS for patients in the trial. In this group, the volume of spectroscopically abnormal tissue (GTV3) was a statistically significant biomarker of both OS and PFS, and performed superior to the extent of resection and residual enhancing tumor. The results of this analysis support using spectroscopic MRI to guide high-dose radiation and suggest that the non-enhancing component of glioblastoma is an important biomarker of survival and should be treated accordingly.

## Figures and Tables

**Figure 1 cancers-15-03524-f001:**
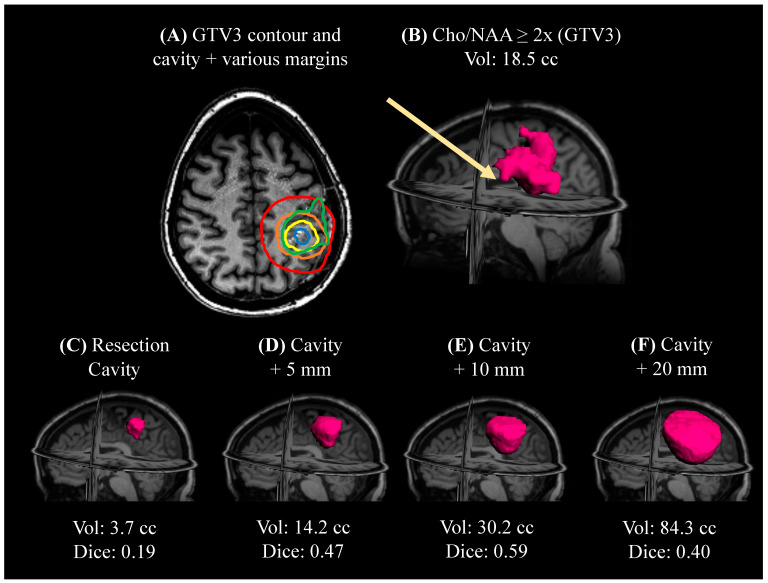
A comparison of an example gross tumor volume 3 (GTV3) treatment volume to the resection cavity with standard of care margins. Due to no residual enhancing tumor, standard of care would involve treating the cavity with a margin. (**A**) An axial view of the GTV3 contour (green) that extends past the resection cavity (blue), resection cavity with 5 mm margin (yellow), 10 mm margin (orange), and 20 mm margin (red). (**B**) This 3D view of the GTV3 contour shows a small amount of infiltrating tumor (arrow) that extends even beyond the 20 mm margin. (**C**–**E**) 3D views of the resection cavity with various margins and volumes are calculated. A Dice overlap score was calculated between each margin and GTV3. For (**C**–**F**), the Dice scores were 0.18, 0.47, 0.59, and 0.40, respectively. The percentages of volumes in (**C**–**F**) not included in (**B**), were 37.4%, 41.3%, 49.7%, and 74.9%, respectively.

**Figure 2 cancers-15-03524-f002:**
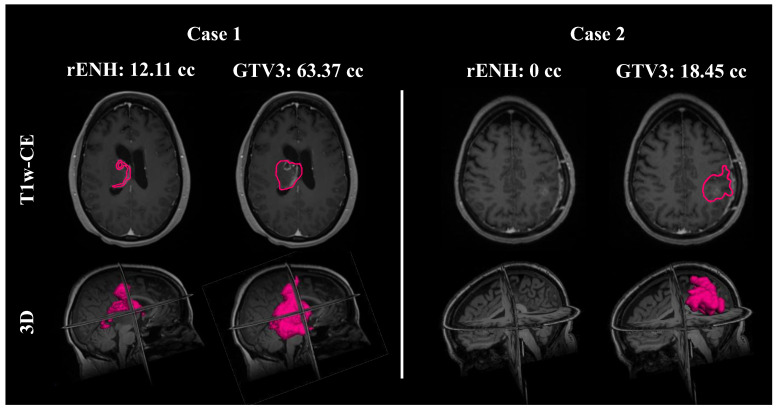
Two sample cases are shown. In the first case, the first column shows the residual post-contrast enhancement (rENH) contour (pink outline) with a volume of 12.11 cc, overlaid on the T1-weighted contrast-enhanced (T1w-CE) MRI (first row) for a 54-year-old female glioblastoma patient. The second column shows the gross tumor volume 3 (GTV3) contour (pink outline), over 5 times larger than rENH with a volume of 63.37 cc. The second row is a three-dimensional rendering of both contour volumes overlaid on a T1 pre-contrast image. In the second case of a 56-year-old female glioblastoma patient, there is no residual contrast enhancement visible on the T1w-CE MRI, whereas the GTV3 volume is 18.45 cc.

**Figure 3 cancers-15-03524-f003:**
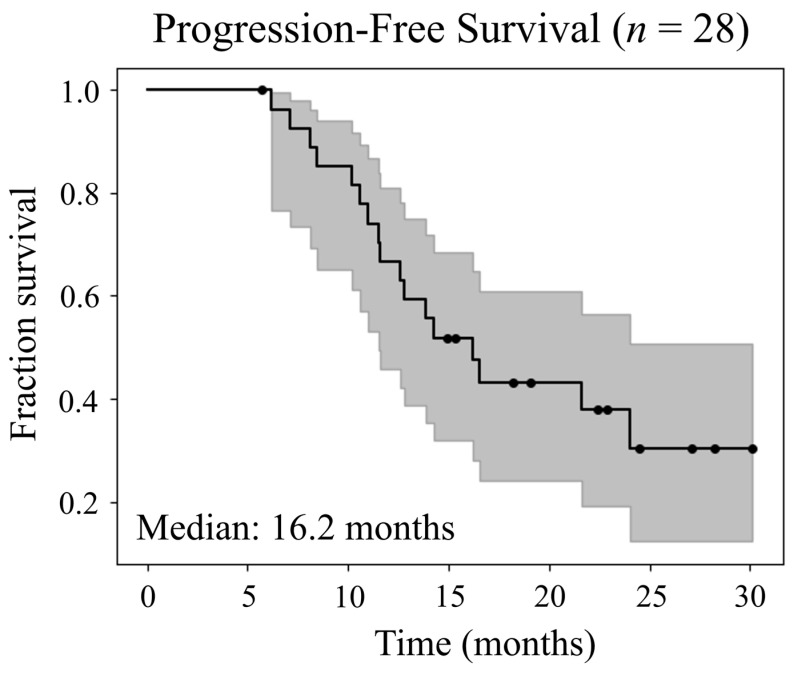
Kaplan–Meier survival curve of progression-free survival (PFS) for the 28 IDH wild-type patients included in this analysis. The median PFS is 16.2 months and median time to follow-up is 22.4 months.

**Figure 4 cancers-15-03524-f004:**
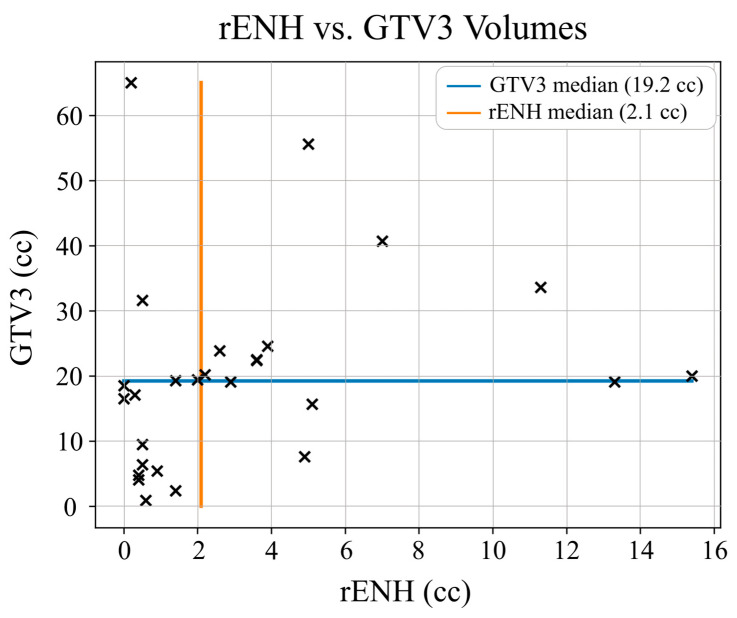
Gross tumor volume 3 (GTV3) and residual post-contrast enhancement (rENH) scatter plot indicating low correlation between the two volumes based on R^2^ = 0.127 (*p* = 0.063). Each marker ‘x’ in the scatter plot corresponds to the GTV3 and rENH volume for each patient in the study. The horizontal blue line indicates the median GTV3 volume (19.2 cc), and the vertical orange line indicates the median rENH volume (2.1 cc).

**Figure 5 cancers-15-03524-f005:**
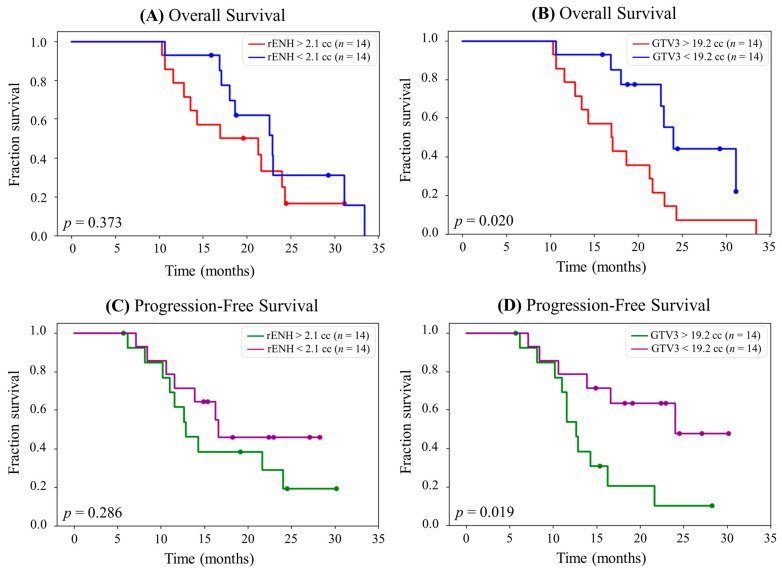
(**A**) Kaplan–Meier survival curve for overall survival (OS) stratified into equal groups by the median residual contrast enhancing (rENH) tumor volume (2.1 cc). The median OS for patients with rENH > 2.1 cc is 21.3 months, and the median for OS for patients with rENH < 2.1 cc is 23.0 months. (**B**) Kaplan–Meier survival curve for OS stratified into equal groups by the median gross tumor volume 3 (GTV3) (19.2 cc). The median OS for patients with GTV3 > 19.2 cc is 17.1 months, and the median OS for patients with GTV3 < 19.2 cc is 27.4 months. (**C**) Kaplan–Meier survival curve for progression-free survival (PFS) stratified into equal groups by the median rENH tumor volume (2.1 cc). The median PFS for patients with rENH > 2.1 cc is 12.8 months, and the median PFS for patients with rENH < 2.1 cc is 19.0 months. (**D**) Kaplan–Meier survival curve for PFS stratified into equal groups by the median GTV3 (19.2 cc). The median PFS for patients with GTV3 > 19.2 cc is 12.6 months, and the median PFS for patients with GTV3 < 19.2 cc is 24.0 months.

**Table 1 cancers-15-03524-t001:** Classification of all 28 patients based on extent of resection (EOR) and residual post-contrast enhancement (rENH) volume. Patients with EOR ≥ 95% had gross total resection (GTR), EOR < 95% had subtotal resection (STR), and the remaining 5 patients had biopsy only. Patients with rENH ≤ 1.5 cc also had GTR, and rENH > 1.5 cc had STR.

	Classification	Number (%) of Patients
EOR	GTR (EOR ≥ 95%)	12 (42.9%)
	STR (EOR < 95%)	11 (39.3%)
rENH	GTR (rENH ≤ 1.5 cc)	13 (46.4%)
	STR (rENH > 1.5 cc)	15 (53.6%)

## Data Availability

No new data were created or analyzed in this study. Data sharing is not applicable to this article.

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
