# Peer review of "Spectroscopic MRI-Based Biomarkers Predict Survival for Newly Diagnosed Glioblastoma in a Clinical Trial"

_cancers, 2023, doi:10.3390/cancers15133524_

Round 1

Reviewer 1 Report

The paper is well and clearly written although there were a few contradictory statements made that are described below.

Line numbers are those listed at right side of each PDF page.

1. Figures 1 and 2. Figure 1 claims to show a rENH volume in blue outline whose volume is non-zero.  Figure 2 claims that Case 2 is the same subject as in Figure 1 but claims to have rENH volume of 0 cc.  This seems contradictory.

2. line 195. In Methods you already explain that you removed the IDH1+ subjects, but here you refer to the '30 patients included in this study'.  Recommend either refer to the 30 subjects in the 'original' study or to the 28 subjects in this study for clarity.

3. line 236. The median OS difference reported for GTV3 here seems to be 23-17.1 = 5.9 months?  Line 234 lists it at 10.3 months. Caption for Figure 5B lists median OS for GTV3 < 19.2 cc as 27.4 months. Please clarify.

Author Response

Reviewer 1:

  1. Figures 1 and 2. Figure 1 claims to show a rENH volume in blue outline whose volume is non-zero.  Figure 2 claims that Case 2 is the same subject as in Figure 1 but claims to have rENH volume of 0 cc.  This seems contradictory.

Thank you for pointing out this error. In Figure 1, the blue outline actually represents the cavity, with the yellow, orange, and red outlines representing the cavity plus a 5, 10, and 20 mm added margin, respectively. In this case where the patient had no residual contrast enhancing tumor, standard-of-care would treat the cavity plus a margin to a high dose of 60 Gy. We have corrected the figure legend and any instances where this is mentioned in the manuscript.

  1. line 195. In Methods you already explain that you removed the IDH1+ subjects, but here you refer to the '30 patients included in this study'.  Recommend either refer to the 30 subjects in the 'original' study or to the 28 subjects in this study for clarity.

We have changed the wording at the beginning of Results (line 195) to instead refer to the 30 patients in the “original study”.

  1. line 236. The median OS difference reported for GTV3 here seems to be 23-17.1 = 5.9 months?  Line 234 lists it at 10.3 months. Caption for Figure 5B lists median OS for GTV3 < 19.2 cc as 27.4 months. Please clarify.

In line 236 the median OS is listed as 10.3 months. According to the caption for Figure 5B, GTV3 < 19.2 cc is 27.4 months and GTV3 > 19.2 cc is 17.1 months. Thus, the calculated difference is 27.4 – 17.4 = 10.3 months.

Reviewer 2 Report

Very interesting study. I have a major concern and a minor observation.

The main problem, as the authors recognize, is the treatment heterogenesis among patients in the cohort. The authors should at least report the treatment for each patient, perhaps in a dedicated table. 

To help the reader throughout the manuscript, results like "The median age for the 28 patients was 59.3 years with a range of 38.1 93 – 71.6 years (minimum – maximum) and there were 10 females and 18 males. Without the 94 two IDH mutant patients, the median OS becomes 21.6[...]" and so forth should appear in the results section rather than in methods.

Author Response

Reviewer 2:

  1. The main problem, as the authors recognize, is the treatment heterogenesis among patients in the cohort. The authors should at least report the treatment for each patient, perhaps in a dedicated table.

The primary treatment heterogeneity was the degree of resection for each patient. While we do have Table 1 in the manuscript which describes the number of gross total resection (GTR) and subtotal resection (STR), we have added a supplemental table including the age, IDH and MGMT status, resection classification (GTR, STR, or biopsy), treatment volume (GTV3), and residual postcontrast enhancement volume (rENH) for each of the 30 patients.

  1. To help the reader throughout the manuscript, results like "The median age for the 28 patients was 59.3 years with a range of 38.1 93 – 71.6 years (minimum – maximum) and there were 10 females and 18 males. Without the 94 two IDH mutant patients, the median OS becomes 21.6[...]" and so forth should appear in the results section rather than in methods.

Thank you for this helpful suggestion to improve the clarity of the manuscript. We have moved that statement and the updated median OS/PFS to the Results.

Reviewer 3 Report

The underlying manuscript offers a lot of data, the authors tried to predicte PFS and OS for patients suffering glioblastoma by GTV3 volume, a spectroscopic MRI-based biomarkers. Although the sample size is small, it is informative for clinical treatment. However, there are still several issues that need to be discussed and revised for the manuscript.

1. Please add references at 52-56.

Introduction:” Despite improvements in the standard of care for glioblastoma patients, it is challenging to identify the extent of tumor infiltration with T1-weighted contrast-enhanced (T1w-CE) MRI and T2-weighted fluid-attenuated inversion recovery (FLAIR) MRI which are currently used to determine the targets of radiation therapy.”

2. The first occurrence of the word needs to be in full.

Materials and Methods (Line 89-90): ”WHO” and ”IDH mutant patients”.

3. In Materials and Methods (Line 117-119), the authors mentioned that ”The gross tumor volume 3 (GTV3) was calculated for each patient by combining the rENH volume with the Cho/NAA ≥ 2x volume to determine the escalated dose target. ”, indicating that GTV3 is not only associated with Cho/NAA ≥ 2x, but also with rENH volume. And in the Results (Line 233-235) , the authors emphasized the importance of GTV3 : "the difference in medians for the groups stratified by GTV3 is 10.3 months (p=0.020), indicating that GTV3 volume is a significant biomarker of OS. ".

However, there is no mention of GTV3 in the abstract (Page 1) , but only Cho/NAA ≥ 2x.

4. Results (Line 221-222):”a scatterplot of rENH vs GTV3 volumes for each patient indicates a low correlation between the volumes (R2=0.127).”

Results (Line 224-225):“Gross tumor volume 3 (GTV3) and residual postcontrast enhancement (rENH) plot indicating no correlation between the two volumes based on R2=0.127. ”

The concept of correlation should be unified. Is it low correlation or no correlation?

5. The authors claim that case 2 is the same patient shown in Figure 1 and Figure 2. However, rENH is labeled as 0cc in Figure 2, while the volume of rENH (blue) is depicted in Figure 1. Please explain the difference about rENH between two figures.

6. In addition to R2, the P value of correlation should be added in Figure 4 or the corresponding text part.

Author Response

Reviewer 3:

  1. Please add references at 52-56. Introduction:” Despite improvements in the standard of care for glioblastoma patients, it is challenging to identify the extent of tumor infiltration with T1-weighted contrast-enhanced (T1w-CE) MRI and T2-weighted fluid-attenuated inversion recovery (FLAIR) MRI which are currently used to determine the targets of radiation therapy.”

We have added a reference for this statement in the introduction.

  1. The first occurrence of the word needs to be in full. Materials and Methods (Line 89-90): ”WHO” and ”IDH mutant patients”.

We have spelled out both acronyms at their first occurrence and used the acronym for all later occurrences.

  1. In Materials and Methods (Line 117-119), the authors mentioned that ”The gross tumor volume 3 (GTV3) was calculated for each patient by combining the rENH volume with the Cho/NAA ≥ 2x volume to determine the escalated dose target. ”, indicating that GTV3 is not only associated with Cho/NAA ≥ 2x, but also with rENH volume. And in the Results (Line 233-235) , the authors emphasized the importance of GTV3 : "the difference in medians for the groups stratified by GTV3 is 10.3 months (p=0.020), indicating that GTV3 volume is a significant biomarker of OS. ". However, there is no mention of GTV3 in the abstract (Page 1) , but only Cho/NAA ≥ 2x.

Thank you for pointing this out. While the GTV3 volume is associated with both Cho/NAA ≥ 2x as well as rENH, in most cases the Cho/NAA ≥ 2x contour completely encompasses the rENH contour. This is further supported by the fact that the correlation between GTV3 volume and rENH volume has an R2 value of 0.127, whereas the correlation between GTV3 volume and Cho/NAA ≥ 2x volume has an R2 value of 0.982. For the abstract, we have re-worded the abstract slightly to indicate GTV3 stratification as “metabolic abnormality” instead of Cho/NAA ≥ 2x because, to your point, GTV3 is not only comprised of the Cho/NAA ≥ 2x volume. We elected to not actually discuss GTV3 to keep the abstract succinct and straightforward.

  1. Results (Line 221-222):”a scatterplot of rENH vs GTV3 volumes for each patient indicates a low correlation between the volumes (R2=0.127).” Results (Line 224-225):“Gross tumor volume 3 (GTV3) and residual postcontrast enhancement (rENH) plot indicating no correlation between the two volumes based on R2=0.127. ” The concept of correlation should be unified. Is it low correlation or no correlation?

Thank you for pointing this out. Since we have a low, but still non-zero R2 value, we have used the term “low correlation” every time these results are mentioned in the manuscript.

  1. The authors claim that case 2 is the same patient shown in Figure 1 and Figure 2. However, rENH is labeled as 0cc in Figure 2, while the volume of rENH (blue) is depicted in Figure 1. Please explain the difference about rENH between two figures.

Thank you for pointing out this error. In Figure 1, the blue outline actually represents the cavity, with the yellow, orange, and red outlines representing the cavity plus a 5, 10, and 20 mm added margin, respectively. In this case where the patient had no residual contrast enhancing tumor, standard-of-care would treat the cavity plus a margin to a high dose of 60 Gy. We have corrected the figure legend and any instances where this is mentioned in the manuscript.

  1. In addition to R2, the P value of correlation should be added in Figure 4 or the corresponding text part.

We have added the p-value of correlation (0.063). This supports the idea that the observed relationship between the variables is purely by chance, indicating no linear relationship.

Round 2

Reviewer 3 Report

None